

# Lack of near-sightedness principle in non-Hermitian systems

Helene Spring[1⋆], Viktor Könye[2], Anton R. Akhmerov[1] and Ion Cosma Fulga[2]

**1** Kavli Institute of Nanoscience, Delft University of Technology,
P.O. Box 4056, 2600 GA Delft, The Netherlands
**2** Institute for Theoretical Solid State Physics, IFW Dresden and
Würzburg-Dresden Cluster of Excellence ct.qmat,
Helmholtzstr. 20, 01069 Dresden, Germany

⋆ helene.spring@outlook.com

## Abstract

The non-Hermitian skin effect is a phenomenon in which an extensive number of states accumulates at the boundaries of a system. It has been associated to nontrivial topology, with nonzero bulk invariants predicting its appearance and its position in real space. Here, we demonstrate that the non-Hermitian skin effect has weaker bulk-edge correspondence than topological insulators: when translation symmetry is broken by a single non-Hermitian impurity, skin modes are depleted at the boundary and accumulate at the impurity site, without changing any bulk invariant. Similarly, a single non-Hermitian impurity may deplete the states from a region of Hermitian bulk.

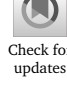

In the absence of long-range interactions, local changes made to an insulator have a local effect. This phenomenon is known as the near-sightedness principle: far from the perturbation, the properties of the system remain as they were [1,2]. Topological insulators, like trivial insulators, obey the near-sightedness principle. The bulk properties of topological insulators stabilize gapless modes at their boundaries in a phenomenon known as bulk-edge correspondence (BEC). symmetry-preserving perturbation at the boundary that destroys the topological phase will locally shift the position of the boundary modes but will not remove them.

In non-Hermitian systems, the near-sightedness principle fails. The spectrum and eigenstates are highly sensitive to boundary conditions: shifting from periodic to open boundary conditions (PBC and OBC) leads to the bulk modes exponentially localizing at the new boundaries [3]. This phenomenon is known as the non-Hermitian skin effect (NHSE). In early works, when the NHSE was discussed from the point of view of non-trivial topology, it was considered to be a failure of the conventional BEC [4]. More recently, it was shown that the 1D NHSE is indeed a topological phenomenon, and the location of the edge modes is predicted by the winding number of the bulk spectrum [5–8]. In higher dimensions however, especially when eigenstate accumulation occurs at corners, multiple invariants have been proposed for different types of NHSE. A recent review has concluded that understanding the formation of corner skin modes is mostly done on a case-by-case basis, and that there is no current consensus on the general theoretical formalism behind it [9].

In the presence of impurities, the failure of the near-sightedness principle in non-Hermitian systems is further demonstrated. Non-Hermitian impurities are observed to attract the modes of the system with a localization length that is proportional to the system size [10–13]. This phase is scale-invariant and is therefore considered distinct from the NHSE phase.

In this work, we show that an appropriately selected non-Hermitian impurity is capable of exponentially localizing all modes present in the system, thus challenging the association between the NHSE and BEC. We show that when translation symmetry is broken, the appearance of this effect as well as its position in real space becomes independent of any bulk topological index. This phenomenon occurs even when the bulk is fully Hermitian, further highlighting the breakdown of bulk-boundary correspondence and the near-sightedness principle. In the following, we explore these features using a simple one-dimensional (1D) model, highlighting first why this effect is expected to occur, followed by a concrete numerical demonstration. We then show this effect is also are present in a two-dimensional (2D) model.

The NHSE can be understood in terms of transfer matrices that relates the wave function at one boundary in a translationally invariant chain to the bulk wave function at a given energy $E$ [14,15]:

$$\begin{pmatrix} \psi(x_{N+1}) \\ \psi(x_N) \end{pmatrix} = T_B^N(E) \begin{pmatrix} \psi(x_1) \\ \psi(x_0) \end{pmatrix}, \tag{1}$$

where $\psi(x_N)$ is the possibly multi-component wave function of the $N$-th unit cell, and $T_B(E)$ is the transfer matrix of one unit cell of the bulk of the chain. In non-Hermitian systems that host the NHSE, there is a preferred direction of transmission towards the boundary with the skin effect. The largest eigenvalue $\lambda_B(E)$ of the transfer matrix $T_B(E)$ representing transmission away from this boundary has a modulus smaller than 1, resulting in the largest eigenvalue of the transfer matrix $T_B^N(E)$ being $|\lambda_B^N(E)| \ll 1$. The magnitude of the eigenvalues of the transfer matrices are therefore directly linked to the accumulation of modes at a certain site: in non-Hermitian systems, they predict which boundary will host the NHSE.

Adding an impurity to the system modifies the transfer matrix. The transfer matrix relating the wave function components on the left side of the chain to those at an impurity on site $N+2$ is given by

$$T(N,E) = T_{\text{imp}}(E) T_B^N(E), \tag{2}$$

where $T_{\text{imp}}(E)$ is the transfer matrix between the wave function components $(\psi(x_N), \psi(x_{N-1}))^T$ and $(\psi(x_{N+1}), \psi(x_N))^T$. If $\lambda_{\text{imp}}(E)$, the smallest eigenvalue of the impurity transfer matrix $T_{\text{imp}}(E)$, is much larger than $\lambda_B^N(E)$, the largest eigenvalue of the bulk transfer matrix $T_B^N(E)$, then all of the modes of the system will accumulate at the impurity site instead of the boundary that hosts the NHSE. Therefore the condition for the NHSE to completely disappear from the system boundary is:

$$\min_E |\lambda_B^N(E)\lambda_{\text{imp}}(E)| \gg 1, \tag{3}$$

where $E$ is any energy that lies within the boundary defined by the PBC eigenvalues of the Hamiltonian—or in other words within the point gap. Eq. (3) describes the case where all of the modes have shifted to the impurity, but the majority of the modes are likely displaced well below this condition. The lower bound on the impurity strength provided by Eq. (3) directly generalizes to the case when the bulk is disordered, in which case slowest decaying eigenvalue of the transfer matrix is replaced by the largest Lyapunov exponent of the system [16]. Alternatively, a weaker but more straightforwardly valid lower bound follows by minimizing $\lambda_B(E)$ over disorder realizations in addition to energy.

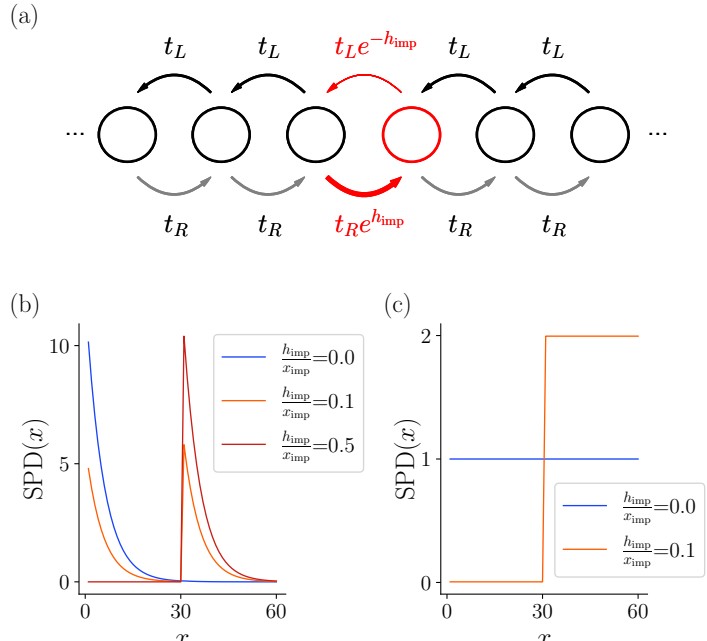

Figure 1: Breakdown of the correspondence of the skin effect and bulk topology via a non-Hermitian hopping impurity in the bulk, model Eq. (4). (a) Schematic of the tight-binding system Eq. (4) around the impurity site (in red). (b) The SPD [Eq. (5)] of a 1D chain of 60 sites in a non-Hermitian system ($t_R = 0.9$ and $t_L = 1.1$) with a non-Hermitian impurity located at $x_{\text{imp}} = 30$, as a function of increasing impurity strength $h_{\text{imp}}$. (c) Same as (b) for a Hermitian system ($t_R = 1$ and $t_L = 1$). Plot details in App. A.

As a concrete example, we now apply our reasoning to the Hatano-Nelson Hamiltonian [17, 18], a 1D single-orbital non-Hermitian Hamiltonian:

$$H(m,N) = \sum_{\substack{j \neq m}}^{N} t_R |j\rangle\langle j-1| + t_L |j-1\rangle\langle j| + e^{h_{\text{imp}}} t_R |m\rangle\langle m-1| + e^{-h_{\text{imp}}} t_L |m-1\rangle\langle m|, \quad (4)$$

where the sum runs over the lattice sites $j$ of the system, $N$ is the total number of sites of the chain, $m$ corresponds to the impurity site, and $h_{\text{imp}}$ models the magnitude of the hopping asymmetry that defines the impurity [Fig. 1 (a)]. $h_{\text{imp}} = 0$ results in a uniform system with no impurity. For simplicity we do not consider onsite terms, and the non-Hermiticity of the bulk arises from the hopping asymmetry in the bulk, $t_R \neq t_L$.

We observe the effect of a non-Hermitian impurity in this model by tracking the spatial distribution of modes in the system, in order to determine its effect on the NHSE. An extensively used method of characterizing the NHSE is the calculation of the real-space sum of probability densities (SPD) of all eigenstates of a system:

$$\text{SPD}(x_j) = \sum_n |\Psi_n(x_j)|^2, \quad (5)$$

where $\Psi_n(x_j)$ is amplitude of the $n$-th eigenvector on site $x_j$. While the local density of states is defined for individual energies, the SPD is akin to a local density of states evaluated at all energies of the system. We set $t_L > t_R$. In doing so, we realize a non-Hermitian system where the NHSE appears on the left of the chain, with modes exponentially localized around site $j = 0$. In non-Hermitian systems, as $h_{\text{imp}}$ increases, the skin effect shifts away from the system

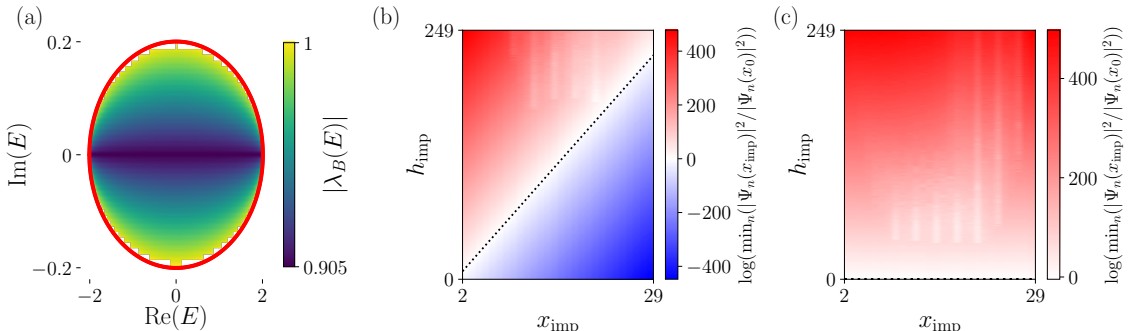

Figure 2: Breakdown of bulk-edge correspondence in a system with an amplifying non-Hermitian impurity, model Eq. (4). (a) The modulus of the largest eigenvalue of $T_B$ [Eq. (6)] for all energies within the boundary defined by the PBC eigenvalues (in red), for $t_R = 0.9$ and $t_L = 1.1$. (b)-(c) the smallest ratio of eigenvector components at the impurity $|\Psi(x_{\mathrm{imp}})|^2$ and the eigenvector components at the left boundary $|\Psi(x_0)|^2$ as a function of the impurity strength $h_{\mathrm{imp}}$ and impurity position $x_{\mathrm{imp}}$ for (b) non-Hermitian ($t_R = 0.0003$ and $t_L = 2980$) and (c) Hermitian systems ($t_R = 1$ and $t_L = 1$). The bound $|\lambda_B^{x_{\mathrm{imp}}-2}(E)\lambda_{\mathrm{imp}}(E)| = 1$ [Eq. (3)] is shown as a dotted line in (b) and (c), where $E$ is the energy for which the modulus of the wave component at the impurity is the smallest. Plot details in App. A.

boundaries to the impurity site in the bulk, as evidenced by the change in SPD [Fig. 1 (b)]. In Hermitian systems ($t_R = t_L = 1$), the non-Hermitian impurity depletes the modes to its left and accumulates them to its right [Fig. 1 (c)].

We now analyze the model Eq. (4) in terms of transfer matrices and the condition Eq. (3). We first examine the transfer matrix of the system without impurities. The transfer matrix relating wave functions of different unit cells in the bulk of the chain is given by:

$$T_B(E) = \begin{pmatrix} E/t_L & -t_R/t_L \\ 1 & 0 \end{pmatrix}. \tag{6}$$

As shown in Fig. 2 (a), the modulus of the largest eigenvalue of $T_B(E)$ [Eq. (6)] is smaller than 1 for any energy that lies within the limits of the PBC spectrum. This means that the largest eigenvalue of the transfer matrix connecting increasingly distant points of the chain will be much smaller than 1.

We now consider the system with an impurity ($h_{\mathrm{imp}} \neq 0$). The transfer matrix relating $(\psi(x_{\mathrm{imp}}), \psi(x_{\mathrm{imp}-1}))^T$ to $(\psi(x_{\mathrm{imp}-1}), \psi(x_{\mathrm{imp}-2}))^T$ is:

$$T_{\mathrm{imp}}(E) = \begin{pmatrix} e^{h_{\mathrm{imp}}} E/t_L & -e^{2h_{\mathrm{imp}}} t_R/t_L \\ 1 & 0 \end{pmatrix}. \tag{7}$$

We diagonalize Eq. (4) for various hopping asymmetry strengths at the impurity located at $x_{\mathrm{imp}}$, and extract the components of all the eigenvectors at the boundary $\Psi_n(x_0)$ and the components at the impurity site $\Psi_n(x_{\mathrm{imp}})$. The smallest ratio of these components

$$\min_n |\Psi_n(x_{\mathrm{imp}})|^2/|\Psi_n(x_0)|^2, \tag{8}$$

belongs to the eigenstate of the system that is the most localized at the boundary. With decreasing impurity distance from the boundary and/or increasing impurity strength, this ratio can be made arbitrarily large [Fig. 2 (b)], indicating that all of the modes of the system accumulate at the impurity for a large enough hopping asymmetry at the impurity.

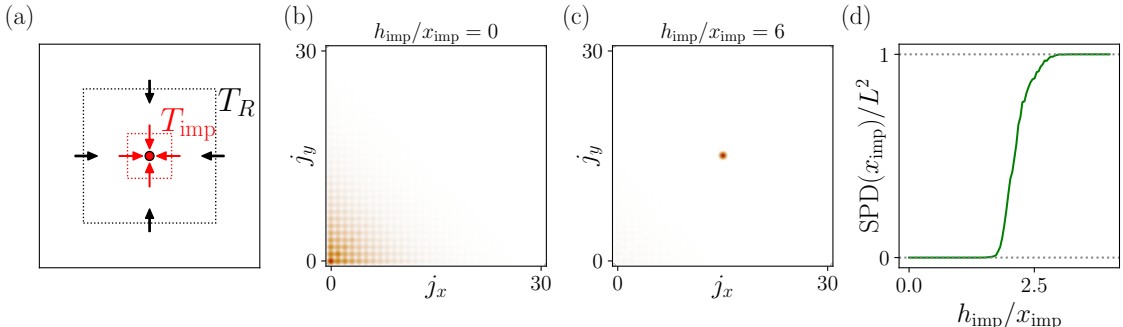

Figure 3: Shifting modes via a non-Hermitian impurity in a 2D non-Hermitian system hosting the NHSE. (a) Schematic of the 2D system with an impurity at the center. Black arrows indicate the direction of transfer operated by the rectangular transfer matrix $T_R$ across the boundary marked by a black dashed line. Red arrows indicate the direction of transfer of the impurity transfer matrix $T_{\text{imp}}$ across the boundary marked by the red dashed line. (b) SPD [Eq. (5)] of a 2D non-Hermitian system Eq. (9) with no impurities. Darker color indicates a larger SPD. (c) SPD of the same bulk non-Hermitian Hamiltonian with an impurity $h_{\text{imp}}/x\text{imp} = 6$. Darker color indicates a larger SPD. (d) SPD at the impurity site as a function of increasing impurity hopping asymmetry $h_{\text{imp}}/x_{\text{imp}}$, in a system with $t_L = t_U = e^1$ and $t_R = t_D = e^{-1}$. Plot details in App. A.

We also calculate $\lambda_B^{x_{\text{imp}}-2}(E)\lambda_{\text{imp}}(E)$, where $E$ is the energy for which the modulus of the wave function at the impurity is the smallest. We use this expression to determine the threshold where the eigenvector most localized at the edge starts to shift towards the impurity, by plotting $\lambda_B^{x_{\text{imp}}-2}(E)\lambda_{\text{imp}}(E) = 1$. As shown in Fig. 2 (b), this threshold aligns with $\min_n |\Psi_n(x_{\text{imp}})|^2/|\Psi_n(x_0)|^2 = 1$, where the most localized eigenstate is equally present at the system boundary and at the impurity. For a fully Hermitian bulk ($t_R = t_L = 1$), the crossover threshold is located at $h_{\text{imp}} = 0$ [Fig. 2 (c)]. Fluctuations in $\min_n |\Psi_n(x_{\text{imp}})|^2/|\Psi_n(x_0)|^2$ present in Fig. 2 (b)-(c) are due to finite-size effects, see App. A.

We now extend our analysis to higher-dimensional systems. In a general $d$-dimensional system, we conjecture that a similar analysis can be performed by examining transfer matrices in the radial direction. We take 2D systems as an example [Fig. 3 (a)]. We consider the following 2D Hamiltonian:

$$
\begin{aligned}
H(m_x, m_y, N_x, N_y) = \sum_{j_x \neq m_x}^{N_x} \sum_{j_y \neq m_y}^{N_y} & t_R |j_x+1, j_y\rangle\langle j_x, j_y| + t_L |j_x, j_y\rangle\langle j_x+1, j_y| \\
& + t_U |j_x, j_y+1\rangle\langle j_x, j_y| + t_D |j_x, j_y\rangle\langle j_x, j_y+1| \\
& + e^{h_{\text{imp}}}(t_L |m_x, m_y\rangle\langle m_x+1, m_y| + t_R |m_x, m_y\rangle\langle m_x-1, m_y| \\
& \quad + t_D |m_x, m_y\rangle\langle m_x, m_y+1| + t_U |m_x, m_y\rangle\langle m_x, m_y-1|) \\
& + e^{-h_{\text{imp}}}(t_L |m_x+1, m_y\rangle\langle m_x, m_y| + t_R |m_x-1, m_y\rangle\langle m_x, m_y| \\
& \quad + t_D |m_x, m_y+1\rangle\langle m_x, m_y| + t_U |m_x, m_y-1\rangle\langle m_x, m_y|),
\end{aligned}
\tag{9}
$$

where the sums run over the coordinate indices of the lattice sites $j_x$, $j_y$ of the system, the impurity is located at $(j_x, j_y) = (m_x, m_y)$, and for simplicity we consider the hopping asymmetry at the impurity $h_{\text{imp}}$ to be the same in both the $x$ and $y$ directions. There are four hopping asymmetry impurities, two to the immediate left and right of the impurity site, and two immediately above and below the impurity site.

In a one-dimensional chain, a transfer matrix argument connecting neighboring sites is sufficient to track the shifting of the modes towards an impurity [Fig. 2 (b)]. In two dimensions, we extend this argument to transfer matrices $T_R(E)$ that connect outer regions of a sample to its inner regions, following the example shown in Fig. 3 (a):

$$\boldsymbol{\psi}_{\text{in}} = T_R(E)\boldsymbol{\psi}_{\text{out}}, \tag{10}$$

where $\boldsymbol{\psi}_{\text{in}}$ are the wave components on the sites that lie immediately within the boundary denoted by the black dashed line, and $\boldsymbol{\psi}_{\text{out}}$ are the wave components on sites lying immediately outside the same boundary. Since the size of $\boldsymbol{\psi}_{\text{in}}$ is smaller than the size of $\boldsymbol{\psi}_{\text{out}}$, $T_R(E)$ is a rectangular matrix.

In the presence of an impurity at the center of a $N \times N$ lattice, the transfer matrix from the outer boundaries to the impurity is given by:

$$T(E) = T_1(E)T_2(E) \cdots T_{N/2-1}(E)T_{\text{imp}}(E), \tag{11}$$

where $T_i(E)$ are rectangular transfer matrices, and $T_{\text{imp}}(E)$ is the impurity transfer matrix as shown schematically in Fig. 3 (a). Since the radial transfer matrices are rectangular, there are wave functions at the edge of the system that inevitably have an exactly zero weight at the impurity. However, wave functions satisfying generic and not fine-tuned boundary conditions have weight in all the components, and therefore have a finite coupling to the impurity. Therefore we expect that in the general case, a non-Hermitian impurity that amplifies wave functions incoming from all directions should suppress all NHSE in a finite sample.

We now verify numerically that a non-Hermitian impurity in 2D is capable of attracting all of the modes in the system. We first consider the system with no impurity ($h_{\text{imp}} = 0$). We set $t_L = t_D = 1.1$ and $t_R = t_U = 0.9$, which results in a NHSE manifesting at the lower-left region of the 2D system [Fig. 3 (b)]. By then increasing $h_{\text{imp}}$, all of the modes of the system are attracted to the impurity [Fig. 3 (c)-(d)]. For Hermitian systems, a similar accumulation of system modes at the impurity site is observed to occur.

We have shown that local non-Hermitian perturbations draw the NHSE into the bulk of a system, which demonstrates the breakdown of BEC of the NHSE in 1D and 2D in the absence of translation symmetry. Predicting the position of the skin effect using topological invariants thus becomes unreliable once translation symmetry is broken. In real/non-ideal systems, translation symmetry is not guaranteed to be preserved, highlighting the importance of studying non-Hermitian systems in a manner that is sensitive to local details, such as wave packet dynamics [19], rather than bulk invariants.

The non-Hermitian impurities that we have considered here affect only a few hoppings, but they not purely local perturbations, in the sense that global information (the system size) is required in order to know how strong the hopping asymmetry at the impurity has to be before attracting all of the modes of the system.

Our work indicates that, owing to lack of a near-sightedness principle, impurities play a much larger role in non-Hermitian systems than they do in Hermitian ones. This may prove useful for experiments seeking to produce a non-Hermitian skin effect in a variety of material and meta-material systems [20–24]. Rather than tailor gain and loss or nonreciprocity throughout the entire bulk of the experimental system, a single, non-Hermitian local perturbation would be sufficient to generate the NHSE.

# Acknowledgments

**Funding information**   A. A. and H. S. were supported by NWO VIDI grant 016.Vidi.189.180 and by the Netherlands Organization for Scientific Research (NWO/OCW) as part of the Frontiers of Nanoscience program. I. C. F. and V. K. acknowledge financial support from the DFG through the Würzburg-Dresden Cluster of Excellence on Complexity and Topology in Quantum Matter - ct.qmat (EXC 2147, project-id 390858490).

**Author contributions**   H. S. made the initial observation of non-Hermitian impurities attracting skin effect modes. H. S. performed the numerical simulations and wrote the manuscript with input from other authors. All authors contributed to the theoretical explanation behind this observation and defined the research plan.

**Data availability**   The data shown in the figures, as well as the code generating all of the data is available at [25].

# A    Model and plotting parameters

In this section additional details of the plots are listed in order of appearance.

For Fig. 1, simulations were done for 1D systems composed of 60 sites. The values of $h_{\text{imp}}$ used are 0, $0.05L$, and $0.25L$, for both the non-Hermitian and the Hermitian systems. For panel (b), the bulk Hamiltonian parameters are $t_L = e^{0.1} = 1.1$ and $t_R = e^{-0.1} = 0.9$. For panel (c), the bulk Hamiltonian parameters are $t_L = 1$ and $t_R = 1$.

For Fig. 2 (a), simulations were done for 1D systems composed of 10 sites. For the non-Hermitian system shown in panel (b), bulk parameters $t_L = e^8$ and $t_R = e^{-8}$ were used. The high hopping asymmetry in the bulk is used to reduce the oscillations of $\min_n |\Psi_n(x_{\text{imp}})|^2/|\Psi_n(x_0)|^2$ that arise due to the penetration of the skin effect into the bulk (as shown for example in Fig. 1 (b)). Parameters $t_L = 1$ and $t_R = 1$ were used for the Hermitian system shown in panel (c).

For Fig. 3, simulations shown in panels (b)-(d) were performed using 2D systems composed of $31 \times 31$ sites with bulk hopping parameters $t_L = t_U = e^1$ and $t_R = t_D = e^{-1}$ (see (9)). In panels (b) and (c), the impurity hopping asymmetry is $h_{\text{imp}}/x_{\text{imp}} = 0$ in (b) and $h_{\text{imp}}/x_{\text{imp}} = 6$ in (c).

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
