# Peer review of "Lack of near-sightedness principle in non-Hermitian systems"

_SciPost Physics, doi:SciPost Phys. 17, 153 (2024)_

## Round 1 · Referee Report · Anonymous (Referee 1) · 2023-12-17

Strengths

The description is clear enough.

Weaknesses

The criticism to all papers on non-Hermitian skin effect is not well supported and unfair.

Report

See attached.

Attachment

  • validity: low
  • significance: poor
  • originality: low
  • clarity: ok
  • formatting: good
  • grammar: good

Author:  Helene Spring  on 2024-08-15  [id 4698]

(in reply to Report 4 on 2023-12-17)

Dear referee,
We attach the full response to both reports, and the redline manuscript with changes highlighted.
Best regards,
The authors

Attachment:

response_and_diff_25YgWzU.pdf

---

## Round 1 · Referee Report · Anonymous (Referee 2) · 2024-1-16

Strengths

  1. The contents of the paper is well organised and clearly presented;
  2. The research presented is timely and interesting;
  3. The authors reveal that a non-Hermitian impurity results in a lack of the near-sightedness principle in otherwise Hermitian systems.

Weaknesses

  1. The observed effect seems somewhat straightforward.

Report

The authors show that the near-sightedness principle breaks down in the presence of a non-Hermitian impurity. They demonstrate this effect for one- and two-dimensional systems both in the case of a non-Hermitian and Hermitian system. While the effect seems somewhat intuitive, I appreciate that the authors put this effect in the wider context of topological systems.

I have a few questions for the authors about their work: 1. The breakdown of the near-sightedness principle is demonstrated for the one-dimensional system in Fig. 1(a). To show the effect, the authors choose t_R= 0.9 and t_L = 1.1, i.e., the states in the model propagate to the left. The impurity is implemented in such a way that if h_imp ≠ 0 and >0, t_L becomes smaller and t_R becomes larger. In other words, at the impurity, the states would prefer to move in the opposite direction as compared to the rest of the chain. As such, it is not surprising to me that turning on h_imp > 0, one would at some point find an h_imp large enough for which all the states will accumulate at the impurity. Did the authors also check whether this effect takes place for h_imp < 0? 2. The text in the paper seems to imply that any non-Hermitian impurity would result in a breakdown of the near-sightedness principle. However, this is probably only the case for a non-reciprocal impurity like the one in red in Fig. 1(a). Is that indeed correct?

Requested changes

  1. The on-site impurity seems to act as if it changes the boundary conditions if its value is large enough. To this end, the authors may wish to cite Roccati, Phys. Rev. A 104, 022215 (2021).
  2. A transfer matrix setup with an impurity has also been studied in Dwivedi, Phys. Rev. B 97, 064201 (2018) for Hermitian systems. I believe this paper deserves a citation.

  • validity: high
  • significance: good
  • originality: good
  • clarity: high
  • formatting: excellent
  • grammar: perfect

Author:  Helene Spring  on 2024-08-15  [id 4697]

(in reply to Report 5 on 2024-01-16)
Category:
answer to question
reply to objection

Dear referee,
We attach the full response to both reports, and the redline manuscript with changes highlighted.
Best regards,
The authors

Attachment:

response_and_diff.pdf

---

## Round 2 · Referee Report · Anonymous (Referee 1) · 2024-9-8

Report

I am satisfied by the change in the abstract, but I am not satisfied by the authors' response to my argument of the imaginary gauge transformation. I did not mean an alternative proof. What I meant was that the present paper complicates a much simpler mathematical fact. I don't think the complication of introducing "impurity" has any physical insight into the phenomenon of non-Hermitian skin effect. I thereby still do not recommend its publication.

Recommendation

Reject

  • validity: low
  • significance: low
  • originality: ok
  • clarity: good
  • formatting: good
  • grammar: good

Author:  Anton Akhmerov  on 2024-09-09  [id 4749]

(in reply to Report 1 on 2024-09-08)
Category:
objection

We are happy that the referee is satisfied with the changes in the abstract. The referee also does not raise any further concerns about the contents of our manuscript, nor do they refute any of the points in our reply.

The referee's assessment of our point regarding the rescaling transformation ignores the explanation in our response. There we say that this rescaling transformation is only applicable to a specific model, while the transfer matrix lower bound applies to all tight-binding nonhermitian models.

The referee also does not substantiate their point about lacking physical insight or their evaluation of the significance of the manuscript as "low". As also confirmed by the other referee, our observation is new in the literature, and the potential breakdown of NHSE is a significant aspect of its analysis.

The referee also evaluates the validity of our manuscript as "low" without providing any support for this evaluation.

For the above reasons we consider the referee report subjective, and not supported by scientific arguments.

---

## Round 2 · Referee Report · Anonymous (Referee 3) · 2024-11-4

Report

I was asked for a quick comment on the dispute between the referee and the authors. My report is as follows:

The Referee Report #4 raises two issues with the manuscript. Firstly, it claims that the "near-sightedness principle" applies to (Hermitian) topological insulators only insofar as the bulk states are localized in real space. This, as the authors point out in their response, is not true, since it applies to band insulators where all the bulk-states are Bloch waves. Even for systems where the localization of all states is forbidden by topology (such as QHE), I would not expect disorder deep in the bulk to destroy the boundary signature without simultaneously destroying the bulk topology. Secondly, the referee uses a non-unitary gauge transformation on the Hatano-Nelson model to essentially arrive at the same result as the manuscript, namely, that a single impurity, if strong enough, can remove all localized modes from the boundary (as is usual for the non-Hermitian skin effect) and localize them on the impurity instead. This, as the authors state in their response, is an instance of their result, whose application is more transparent in general owing to its use of transfer matrices.

In my opinion, while one may certainly debate the centrality of a nearsigntedness principle to a topological insulator, I think the authors have certainly demonstrated that it does not apply to the non-Hermitian skin effect, and that the boundary modes arising from it act quite unlike the topological boundary modes of Hermitian topological insulators. In conclusion, I do not believe that Referee Report #4 provides sufficient grounds for rejection of this manuscript.

Recommendation

Publish (meets expectations and criteria for this Journal)

---

## Round 2 · Author Response

Dear editor,

We have now responded to the referees and attached a redlined version of the manuscript with marked changes compared to the previous submission to our responses.

We believe that we have addressed the main concerns expressed in report 5 by clarifying the manuscript and addressing some of the misunderstanding.

Best regards,
The authors

---

## Round 2 · List of Changes

Listed in full in the replies to referees of the first submission.

---

## Editorial Decision

published